# Modelling Faecal Sludge Dewatering Processes in Drying Beds Based on the Results from Tete, Mozambique

**Osvaldo Moiambo \***, **Raúl Mutevuie, Filipa Ferreira** and **José Matos**

CERIS, Instituto Superior Técnico, Universidade de Lisboa, Av. Rovisco Pais 1, 1049-001 Lisboa, Portugal;
bombatejunior@gmail.com (R.M.); filipamferreira@tecnico.ulisboa.pt (F.F.);
jose.saldanha.matos@tecnico.ulisboa.pt (J.M.)
\* Correspondence: moiambo03@gmail.com; Tel.: +258-848-856-078

**Abstract:** Currently, in sub-Saharan African countries, 65 to 100% of the urban population is served by on-site sanitation systems, typically pit latrines and septic tanks. Faecal sludge dewatering in drying beds is one of the most commonly adopted low-cost treatment technologies in developing countries due to favourable climate conditions in terms of temperature and radiation and because their operation and maintenance are simple. Nevertheless, research in tropical dry zones regarding these processes is scarce. In this paper, a mathematical model for faecal sludge dewatering in sludge drying beds (SDBs) focusing on gravity drainage and evaporation processes is presented and discussed. Experimental campaigns were carried out at a pilot site in Tete, Mozambique, to determine the model parameters. The results suggest that the model can be used to estimate, in similar situations, the dewatering process of different types of faecal sludge. The present research might be considered as a contribution to the design and operation of SDBs, supporting sludge management and allowing the estimation of drying times and optimal loading cycles, namely, the sludge thickness and final moisture content.

**Keywords:** dewatering; evaporation; faecal sludge; sludge drying beds

## 1. Introduction

In developing countries, the urbanisation process was not accompanied by a proportional expansion of adequate sanitation infrastructures or by the rehabilitation of old infrastructures that aged over time. Sanitation conditions have become especially critical in peri-urban areas where proper infrastructures and services are usually almost completely non-existent. Currently, access to sanitation in these peri-urban areas is typically ensured by on-site sanitation techniques, namely, pit latrines and septic tanks. In sub-Saharan Africa, for example, about 65 to 100% of urban areas are served by these types of systems [1].

The expansion of sanitation decentralised systems is characterised by an increase in the amount of sludge accumulated in these structures, particularly the faecal sludge, a mixture of excreta and water, which has a variable consistency accordingly to the degree of digestion [2]. As investment in faecal sludge management in developing countries has increased, there is an urgent need to adopt strategies that allow adequate sludge management either by developing and rehabilitating existing systems or by investing in new low-cost solutions that fit the local reality. In particular, it is important to implement technologies that provide a significant reduction in the volume of sludge and, consequently, a reduction in costs associated with handling, transportation, treatment and final disposal operations.

Sludge drying beds (SDBs) are notable for being of a low cost and based on natural processes, namely, evaporation and gravitational drainage [3]. SDBs are simple and economical solutions, providing a significant reduction in the volume of sludge and improvements in terms of sludge hygiene and structure (e.g., [4–6]).

The performance of SDBs is related to the duration of the drying cycle and depends, fundamentally, on the local climate and on the sludge characteristics, namely, the solid

content and degree of stabilisation. However, knowledge and experience about these dewatering technologies are still incipient, with few studies regarding the factors that most influence the dewatering process. Research is particularly scarce when it comes to sub-Saharan Africa and the specialised literature proposes different ranges of values for the design criteria deduced for other parts of the world.

The mathematical modelling of SDBs can be useful for an optimised design and operation and provide a better understanding of the factors that may affect the performance of the drying beds. In addition, adequate modelling provides useful information on the response of an installation to different environmental scenarios.

One of the first mathematical models of the sludge dewatering process in drying beds was developed over 50 years ago by Lo [7]. At that time, modelling was only limited to the estimation of the area required to dehydrate the sludge. Another integrated approach was proposed in 2001 in Brazil [5], comprising both evaporation and drainage processes. However, most authors (e.g., [8–12]) opted for a partial modelling approach considering only one component of the dehydration process (evaporation or drainage). In addition, few SDB models have been developed and calibrated with the data of sub-Saharan countries where it is believed that this type of technology may become more widely applied.

In the present paper, we refer to the development of an integrated SDB mathematical model that aims to predict the retention times required for faecal sludge dewatering in order to reach a certain solid content or dry matter percentage based on the application of equations that describe the water balances. Experimental campaigns were carried out at a pilot site in Tete, Mozambique, to obtain data for the calibration of the proposed model. This model may be applied in practice to better operate existing drying bed systems in view of the climate and other local variables or to support the planning of new SDB systems in developing countries of sub-Saharan Africa.

## 2. Materials and Methods

### 2.1. Model Development

2.1.1. Theoretical Overview of the Sludge Dewatering Processes of Drying Beds

The sludge dewatering process in drying beds is based on the solid–liquid separation by two mechanisms: drainage and evaporation. For a better understanding of the mathematical model, this section presents a brief theoretical overview of the removal mechanisms.

Drainage Process

Drainage is a filtration process in which the filter medium retains a fraction of the solids at the surface and allows the liquid mass to pass through [13,14]. When the SDBs are loaded, three distinct phases may occur (Figure 1): (a) an initial phase, during which the suspended solids deposit on the surface of the filter medium with formation of the "cake", leaving a lesser dense liquid mass interface at the top; (b) an intermediate phase, or filtration phase, in which the thickness of the cake remains approximately constant and the depth of the liquid at the top progressively decreases by drainage; (c) the final phase, which is characterised by the disappearance of the liquid mass from the cake and the formation of cracks within the solid material. The drainage process ceases for two main reasons: the reduction of the hydraulic load and an increase in cake resistance.

Considering that pressure losses in pipes and in the filter medium are relatively small, the resistance to flow may be considered to result solely from the cake resistance. The cake specific resistance depends mainly on the initial thickness of the sludge layer and the initial concentration of solids and varies along the sludge layer thickness, largely due to compressibility [7,10,15].

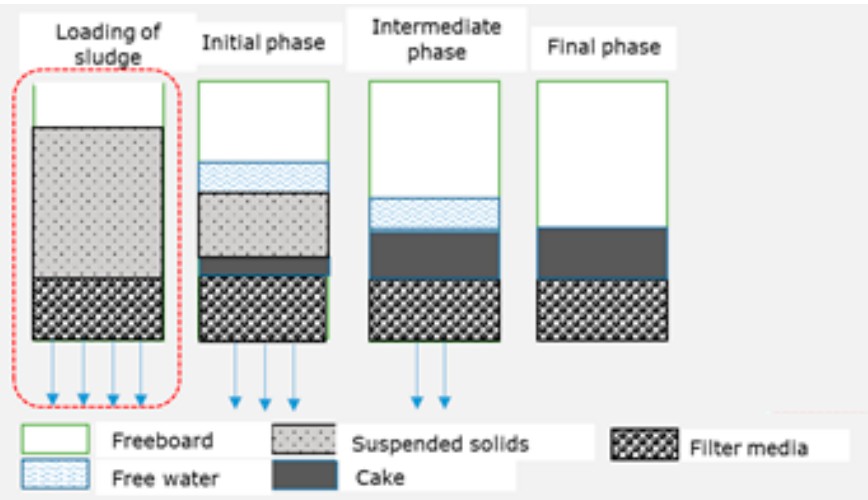

**Figure 1.** Schematic presentation of the phases that characterise the SDB drainage process.

Evaporation

The evaporation of the sludge liquid mass involves an air–water interaction and is controlled by heat and mass transfer phenomena in a transient state. These occur simultaneously with kinetic processes, such as the reduction of the sludge thickness, that can change the initial properties of the sludge (e.g., colour, texture, porosity, odour), affecting the initial mass and energy transfer mechanisms over time [16,17].

When the drying beds are loaded, the transfer of energy occurs in the form of heat from the surrounding medium to the sludge, resulting in the evaporation of the surface water mass. There is also an internal transfer of water to the surface and its subsequent evaporation in response to the previous process. The first process can occur by conduction, radiation and/or convection and depends on the local conditions, e.g., temperature, humidity, wind and the surface area of the drying beds, which are particularly important during the initial phase of evaporation [17,18].

The evaporation process, represented in Figure 2, includes three distinct phases after the loading operation: (i) a brief initial phase (section A-B or A'-B), during which the evaporation rate increases or decreases according to the temperature of the sludge [19]; (ii) an intermediate phase (section B-C) in which the transfer of moisture from the sludge to the free surface is sufficient to keep it completely moist, allowing evaporation at a constant rate where the duration of this phase depends on the sludge surface moisture content and on the amount of free water in the sludge [7,20]; (iii) a final phase (section C-D-E) of a decreasing evaporation rate that begins when the sludge free surface water starts to decrease despite the increase in the surface layer temperature. Along section C-D, the drying curve shows a generally linear progression as the resistance to the internal diffusion of the liquid is small compared to the resistance to the surface vapour removal [19]. Along section D-E, evaporation occurs inside the slurry and the pores fill with air so that the water content decreases until reaching point E, the equilibrium moisture content, which depends on the air temperature and humidity [7].

2.1.2. Model Scope and Key Assumptions

An SDB schematic layout is shown in Figure 3. The sludge surface $A_s$ is exposed to local climate conditions and the heat exchange may occur with the outside environment through the control volume, e.g., volume V, corresponding with the applied sludge layer $h$. The SDBs are loaded in a discontinuous operating regime.

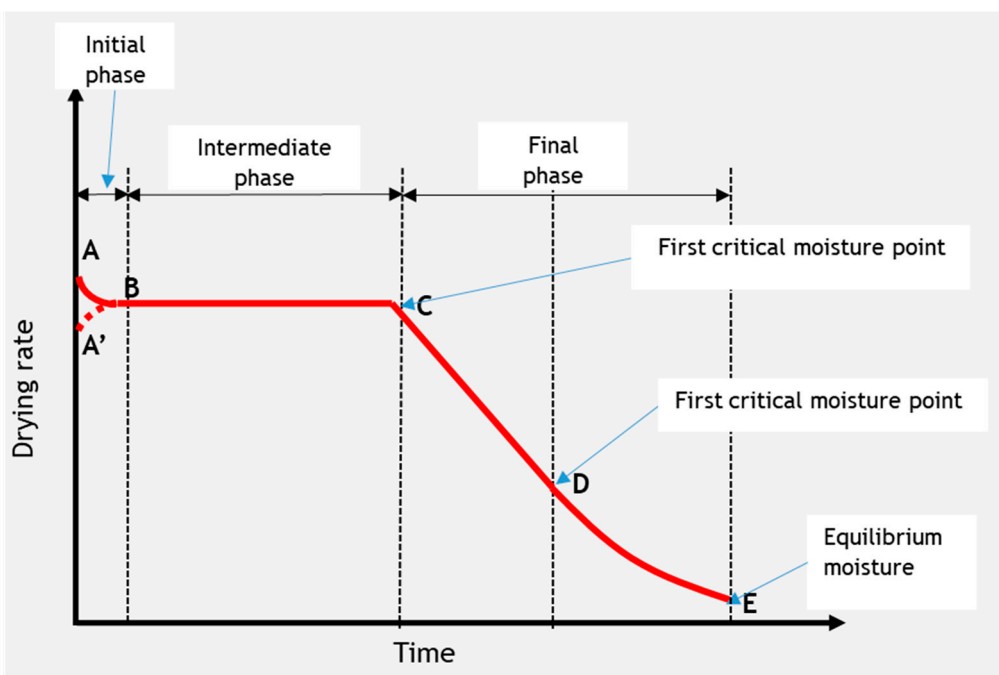

**Figure 2.** Schematic characterisation of the different phases of sludge dewatering at drying beds.

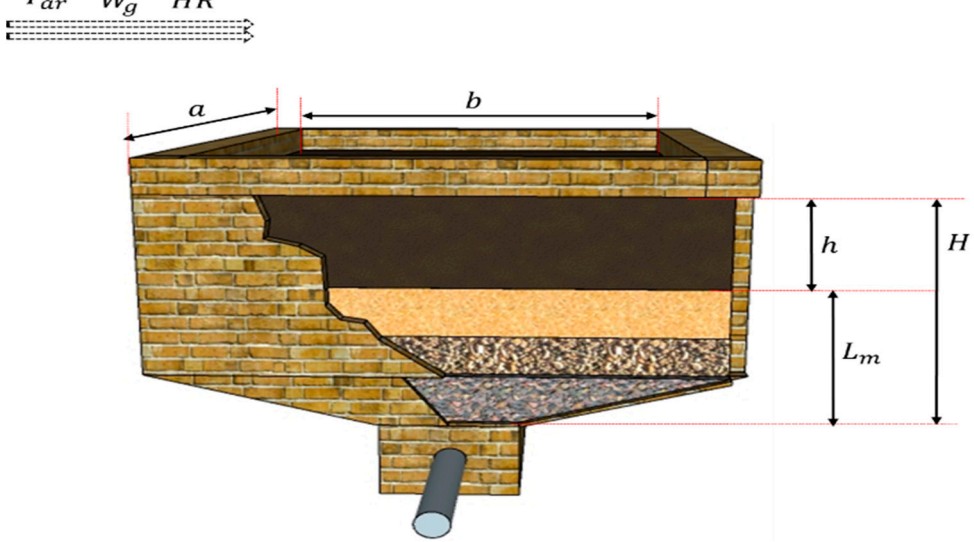

**Figure 3.** SDB schematic layout.

The model development considered the following key assumptions:

- Sludge is considered heterogeneous, with solid and fluid phases. The water mass in the sludge is present in free, interstitial, adsorbed and intracellular fractions.
- Air is considered to be an ideal gas with a constant composition that flows perpendicularly to the SDB with a certain temperature ($T_{air}$), wind speed ($\omega_g$) and relative humidity ($HR$).
- Drying beds are subject to atmospheric pressure and temperature.
- During SDB loading, the influence of the settling velocity on the drainage process is neglectable.
- Sludge solids are deposited over the entire surface of the filter medium, forming a homogeneous porous layer with a constant permeability.

- The effect of biochemical transformations that may occur throughout the dewatering process is neglected both in the sludge and inside the filter medium.
- Drainage processes start after the cake is formed, corresponding with the intermediate phase.
- The energy available for the drainage is constant.
- The mass of solids remains constant as evaporation occurs.
- Evaporation occurs from the interfacial area of the sludge layer, neglecting the transfer of mass and energy through the side walls and through the bottom of the drying beds.

As shown in Figure 3, the model has different boundaries: the upper part of the sludge drying beds is in contact with the atmosphere and local weather conditions, the bottom drainage materialises from a pipe with a free discharge to the atmosphere (being the discharged effluent collected in a container) and the lateral walls, where no hydraulic mass exchange takes place. On the surface of the SDB, it is assumed that the model receives water contained in the sludge and from precipitation events, losing water through evaporation; at the bottom, water is lost through gravitational drainage and its control is carried out by the installed short pipe, which includes a sectioning valve.

2.1.3. Mathematical Formulation of the Sludge Dewatering Process

The mathematical model is based on water and solid balances in the SDB, considering the approach proposed by [11,12].

The dewatering process in SDBs includes a set of variables that reflect losses and gains of moisture as a function of time to establish an equilibrium. In this way, the general equation that translates the moisture content of the sludge in a given time, $\frac{dTH}{dt}$, is Equation (1) where $m_S$ and $m_w$ are the mass of the solid and liquid fractions present in the sludge, respectively (kg).

$$\frac{dTH}{dt} = \frac{m_w}{m_w + m_S}. \tag{1}$$

The mass of the solid fraction ($m_S$) can be determined according to Equation (2), which represents the amount of solids present in the sludge in an instant (t) in view of the surface area (A), the mean thickness of the sludge mass ($h_0$), the density of the sludge to be dehydrated ($\rho_w$) and the solid content of the sludge ($TS_0$).

$$m_{S(t)} = A \times h_0 \times TS_0 \times \rho_w. \tag{2}$$

The variable $m_w$ can be determined from Equation (3), which represents the mass balance in the system at a time (t) where $m_{w0}$ corresponds with the initial water mass fraction in the sludge (kg), $m_P$ corresponds with the mass of water precipitated on the volume control (kg) and $m_d$ and $m_{ev}$ correspond with the drained and evaporated water (kg), respectively.

$$m_{w(t)} = m_{w0} + m_P - m_d - m_{ev}. \tag{3}$$

Initial Water Mass

The initial water mass present in the volume control, $m_{w_0}$, when the sludge is discharged can be determined according to Equation (4) where $\rho_w$ represents the density of the sludge to be dehydrated and $TH_0$ is the initial moisture content.

$$m_{w_0} = A \times h_0 \times TH_0 \times \rho_w. \tag{4}$$

Precipitated Water Mass

The mass of water precipitated on the control volume ($m_{P(t)}$) over a given time interval ($h_P$) is obtained through Equation (5), considering the rainfall depth in the time interval ($h_P$).

$$m_{P(t)} = A \times h_P \times \rho_w. \tag{5}$$

Drained Water Mass

The total drained water ($m_d$) can be directly measured in field tests or directly determined from Equation (3) in periods without rainfall if the contribution of evaporation is neglected. According to [12], the variable $m_{d_{(t)}}$ can be determined from Equation (6), reflecting the evolution of the water mass in the sludge as the drainage process occurs until the field capacity of the sludge is achieved (e.g., the moisture content of the sludge after the drainage ceases):

$$m_{d_{(t)}} = m_{C_c} + (m_{w_0} - m_{C_c})e^{\left[\frac{(K_d \times t)}{(t - t_d)}\right]} \tag{6}$$

where $m_{C_c}$ represents the water mass in the sludge when the drainage ceases, $t_d$ is the total drainage time, t is the time and $K_d$ is an empirical constant reflecting the resistance provided by the cake and the filter medium to the water flow.

The parameter $m_{C_c}$ can be estimated from the initial mass of solids in the sludge and the sludge field capacity ($TH_{C_c}$) by Equation (7).

$$m_{C_c} = \frac{m_s \times TH_{C_c}}{(100 - TH_{C_c})}. \tag{7}$$

The value of $TH_{C_c}$ depends on several factors including the sludge characteristics, namely, the concentration of solids, the organic matter content, the particle size and the interstitial fractions of the sludge. $TH_{C_c}$ can be obtained from Equation (8) where C represents the initial concentration of solids and $k_{C_c}$ and $\alpha_{C_c}$ are the calibration parameters.

$$TH_{C_c} = k_{C_c}C^{-\alpha_{C_c}}. \tag{8}$$

The drainage resistance, $K_d$, can be determined from Equation (9), according to [21]. In Equation (9), the sum of the cake resistance, $R_c$, and the filter medium resistance, $R_m$, represents the total resistance to water drainage; g is the acceleration of gravity and $\mu$ is the water dynamic viscosity.

$$K_d = \frac{\rho_w g}{\mu(R_c + R_m)}. \tag{9}$$

$R_c$ can be determined from Equation (10) where $\alpha$ represents the specific resistance of the cake and c the initial thickness of the sludge mass, $h_o$ [22–24].

$$R_c = \alpha.c.h_o. \tag{10}$$

$R_m$ can be estimated from Equation (11) where $K_m$ represents the permeability of the filter medium, K is the saturated hydraulic conductivity and $L_m$ is the depth of the filter medium.

$$R_m = \frac{L_m}{K_m} = \frac{L_m \rho_w g}{\mu K}. \tag{11}$$

The total drainage time, $t_d$, depends on the initial thickness of the sludge mass, $h_o$, and can be given by Equation (12) where $k_{t_d}$ and $\alpha_{t_d}$ are calibration parameters dependent on the sludge characteristics.

$$t_d = k_{t_d}h_0{}^{\alpha_{t_d}}. \tag{12}$$

Evaporated Water Mass

The water transferred from the liquid laminar film to the surrounding air ($m_{ev}$) over the interfacial area of the sludge ($A_s$) depends on the mass transfer coefficient ($k_x$) and the concentration gradient between the air humidity on the sludge surface ($\mathcal{X}_s$) and on the humidity of the surrounding air ($\mathcal{X}_f$). It can be estimated by Equation (13) [11] where $F_1$ and $F_2$ represent the correction factors proposed by [11].

$$m_{ev} = k_x(\mathcal{X}_s - \mathcal{X}_f)A_sF_1F_2. \tag{13}$$

$K_x$ can be estimated from Equation (14), according to [11], and represents the mass transfer between a particle and a gas that flows at a given velocity using dimensionless numbers, the Sherwood number (Sh), the Reynolds number (Re) and the Smith number (Sc). In Equation (14), D corresponds with the diffusivity of water vapor in the air, $M_a$ with the molar mass of the air, P with the total pressure and b with the width of the drying bed. R is the ideal gas constant and T the absolute temperature.

$$K_x = \frac{D \, Sh \, M_a P}{b \, R \, T}. \tag{14}$$

The diffusion of water vapor in the air at low temperatures [0–100 °C] may be estimated through the empirical correlation established in [25], Equation (15). The temperature value, T, corresponds with the arithmetic mean of the temperature of the sludge and the surrounding air, Equation (16) [25].

$$D = 1.97 \times 10^{-5} \times \frac{1.01325 \times 10^5}{P} \left( \frac{T}{256K} \right)^{1.685}. \tag{15}$$

$$T = \frac{T_{air} + T_{sludge}}{2}. \tag{16}$$

The air molar mass, $M_a$, can be estimated by Dalton's law [25] from Equation (17) where HR represents the relative humidity of the air, $P_{sat}(T_{air})$ the water vapor pressure at the air temperature and $M_{dry\ air}$ and $M_w$ the molar mass of the dry air and water, respectively.

$$M_a = M_{dry\ air} \left[ \frac{P - (HR \times P_{sat}(T_{air}))}{P} \right] + M_w \left[ \frac{HR \times P_{sat}(T_{air})}{P} \right]. \tag{17}$$

The average Sherwood number, $S_h$, (Equations (18) and (19)), which describes the ratio of convective and diffusive mass transport, can be determined by correlations with a parallel flow over a plate, depending on the flow regime (turbulent or laminar), expressed by the Reynolds number (Equation (21)) and the Schmidt number (Equation (20)) [18]:

(a)   turbulent regime ($5 \times 10^5 < R_e < \infty; 0.8 < S_c < \infty$):

$$S_h = \frac{0.037 S_c R_e^{0.8}}{1 + 2.44 R_e^{-0.1} \left( S_c^{2/3} - 1 \right)}. \tag{18}$$

(b)   laminar regime ($0 < R_e \leq 5 \times 10^5; 0 < S_c < \infty$):

$$S_h = 0.8 (R_e S_c)^{0.1} + \frac{1.47}{\left[ 1 + \left( 1.67 S_c^{1/6} \right)^2 \right]^{1/2}} \times \frac{R_e S_c}{\left[ 1 + 1.30 (R_e S_c)^{1/2} \right]}. \tag{19}$$

$$S_c = \frac{\vartheta_g}{D}. \tag{20}$$

$$R_e = \frac{\omega_g \times b}{\vartheta_g}. \tag{21}$$

Air viscosity, $\vartheta_g$, in the boundary layer can be determined by the linear interpolation of the experimental values for dry air, supplied by [26] at an atmospheric pressure in the temperature range of 0 to 50 °C, using Equation (22):

$$\vartheta_g = [135 + 0.904 \, (T - 273.15)] \times 10^{-7}. \tag{22}$$

Assuming that both laminar films near the air–water interface are saturated, $\mathcal{X}_s$ and $\mathcal{X}_f$ can be easily estimated from Equations (23) and (24), according to [11], where $P_{sat}(T_{air})$ and

$P_{sat}\left(T_{sludge}\right)$ are, respectively, the partial pressure of the water vapor in the atmosphere and on the surface of the sludge. $\mathcal{X}_f$ and $\mathcal{X}_s$ may be estimated from the classic Antoine equation, Equation (25).

$$\mathcal{X}_f = \frac{HR \times P_{sat}(T_{air}) \times M_{water}}{P - HR \times P_{sat}(T_{air}) \times M_{dry\ air}}. \tag{23}$$

$$\mathcal{X}_s = \frac{P_{sat}\left(T_{sludge}\right) \times M_{water}}{P - P_{sat}\left(T_{sludge}\right) \times M_{dry\ air}}. \tag{24}$$

$$P_{sat}\left[\left(T_{sludge}\right),\left(T_{air}\right)\right] = 10^{A_1 - \frac{B_1}{T+C_1}}. \tag{25}$$

The formation of a layer of dry sludge, the "skin", near the sludge interface that tends to dry quicker limits the incidence of solar radiation in the lower layers, thus reducing evaporation rates. The correction factor $F_1$ expresses this fact and may be estimated from Equation (26) [11] where $TH_{(BS)}$ is the moisture content of the sludge, $TH_{0\_(BS)}$ is the initial moisture content and $TH_{e\_(BS)}$ is the equilibrium moisture content with the empirical parameter $n_1$ reflecting the effect of the interfacial layer.

$$F_1 = \left(\frac{TH_{(BS)} - TH_{e\_(BS)}}{TH_{0\_(BS)} - TH_{e\_(BS)}}\right)^{n_1}. \tag{26}$$

The correction factor $F_2$ reflects the formation of cracks on the sludge surface as a result of the sludge retraction. Under these circumstances, the lower layers will be more exposed to the environment conditions, favouring the evaporation process. According to [11], Equation (27) takes this effect into account, assuming that the product $k_x A_s$ in Equation (13) can decrease up to 30% as the evaporation occurs.

$$F_2 = 1 - 0.3\left(\frac{TH_{0\_(BS)} - TH_{(BS)}}{TH_{0\_(BS)} - TH_{e\_(BS)}}\right). \tag{27}$$

The equilibrium moisture content, $TH_{e\_(BS)}$, depends on the nature of the sludge and on the air humidity and temperature. It can be determined through Equation (28) and Equation (29) [11] where $TH_{m\_(BS)}$ represents the moisture content of the sludge layer and $C_2$ is an empirical parameter.

$$TH_{e\_(BS)} = TH_{m\_(BS)}\left[\frac{HR \times K_e \times C_2}{(1 - HR \times K_e) \times (1 - HR \times K_e(1 - C_2))}\right]. \tag{28}$$

$$K_e = 0.84 \exp\left[3366\left(\frac{1}{T_{ar}} - \frac{1}{303.15}\right)\right]. \tag{29}$$

### 2.2. Pilot Facility

A pilot facility was installed in Tete, Mozambique, at the Zambezi Regional Water Authority facilities, to provide data to estimate the SDB model parameters based on the local conditions. Temperature, relative air humidity, wind speed, solar insolation and precipitation data were provided by the local National Meteorological Institute.

The pilot facility included three SDB units (designated LS01, LS02 and LS03) made of plastic; each one had 1 m² of area and was installed on a brick base at approximately 0.5 m from the ground (Figure 4). This setting avoided contact with the ground and the significant heat changes that could result from it. In addition, it allowed the installation of a plastic container underneath the SDB to collect and quantify the drained water.

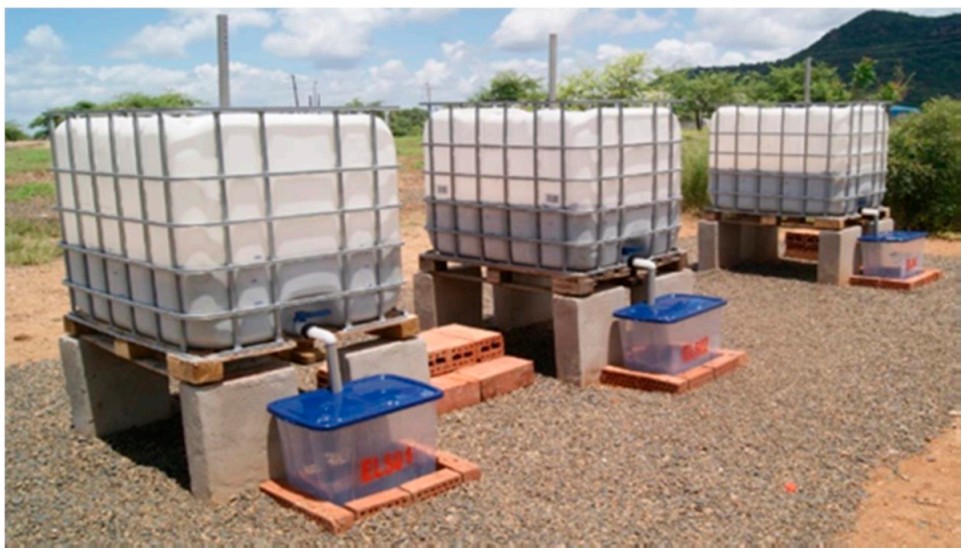

**Figure 4.** Photo of SDB pilot units at Tete, Mozambique.

Each drying bed had filter media of 0.4 m in depth, composed of sand (a top 0.10 m layer with an effective size between 0.3 and 0.6 mm) and gravel (a bottom 0.20 m layer with an effective size of 19 mm and an intermediate 0.10 m layer with an effective size of 10 mm), with a drain system at the bottom. A vertical perforated PVC tube was also installed inside each bed, operating as a piezometer.

Prior to the tests, the initial saturated hydraulic conductivity (K) of the filter medium was determined. The experimental procedure to evaluate the hydraulic conductivity consisted of filling each of the beds with water until a certain level, N (0.47, 0.55, 0.60, 0.65, 0.80, 0.85 and 0.90 m) was reached and then raising the discharge pipe to a certain height, H (0, 0.25 and 0.44 m) below level N. The measurement of the discharged flow, at a constant load, was performed with the recording of the values of Q, N and H. The value of the hydraulic conductivity was estimated from Equation (30) where K is the saturated hydraulic conductivity (m/s), $L_m$ is the thickness of the filter medium (m), h is the height difference between the constant level N and the discharge height H (m), A is the bed section area (m²), Q is the flow rate (m³/s), V is the water volume (m³) and t is the time elapsed for filling a given volume (s). The experimental procedure is described in detail in [27]. The results indicated a mean K value of $4 \times 10^{-4}$ m/s for the three SDB units.

$$K = \frac{QL_m}{hA} = \frac{Vt^{-1}L_m}{(N-H)A}. \tag{30}$$

## 3. Experimental Results

The experimental campaigns were carried out firstly between May and August 2017. The source of the faecal sludge to load the SDB was septage from the septic tanks of households (ST) and public toilets (PT) and was collected and transported in a municipal truck made available by the Tete municipality. Figure 5 illustrates loading of the pilot units.

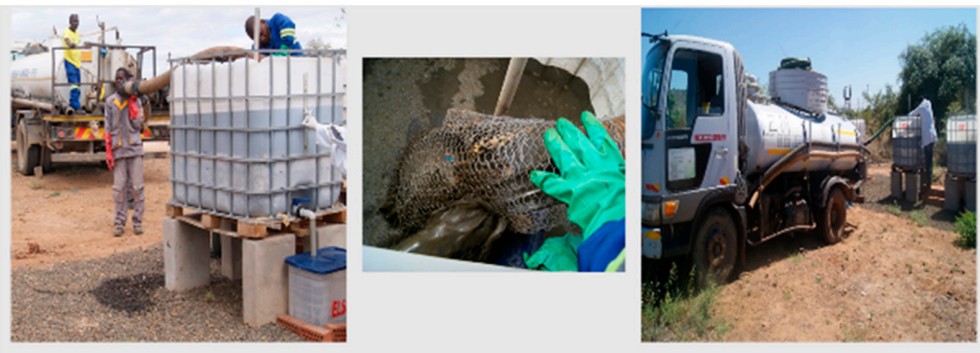

**Figure 5.** Photos of loading the SDB pilot units.

Table 1 shows the experimental data referring to the experimental campaigns.

**Table 1.** Experimental campaign characteristics.

| Period | Cycle | SDB Ident. | Test Ident | Sludge Source | Initial Sludge Depth (m) |
|---|---|---|---|---|---|
| 9 May 2017 to 7 June 2017 | 1st | LS01 | ENS01 | ST01 | 0.20 |
| | | LS02 | ENS02 | ST01 | 0.35 |
| | | LS03 | ENS03 | ST01 + PT01 | 0.56 |
| 22 June 2017 to 10 July 2017 | 2nd | LS01 | ENS04 | ST02 | 0.30 |
| 1 August 2017 to 17 August 2017 | 3rd | LS01 | ENS05 | ST03 | 0.30 |
| | | LS03 | ENS06 | ST03 + PT01 | 0.32 |

Table 2 presents the septage characteristics and average climate data during the experimental campaigns.

**Table 2.** Experimental campaign climate data and septage characteristics.

| Variable | ENS01 | ENS02 | ENS03 | ENS04 | ENS05 | ENS06 |
|---|---|---|---|---|---|---|
| Sludge type | ST | ST | ST + PT | ST | ST | ST + PT |
| Loading date | 9 May 2017 | 9 May 2017 | 9 May 2017 | 22 June 2017 | 1 August 2017 | 1 August 2017 |
| Conductivity (µS/cm) | 1528 | 1528 | 1959 | 908 | 3790 | 2740 |
| Average sludge temperature (°C) | 28.8 | 28.8 | 28.5 | 27.2 | 28.0 | 23.5 |
| pH | 7.8 | 7.8 | 7.5 | 6.8 | 7.6 | 8.0 |
| Initial solid content (%) | 5.2 | 5.2 | 4.0 | 5.4 | 2.0 | 2.0 |
| Air average temperature (°C) | 26.3 | 26.2 | 26.0 | 24.1 | 24.5 | 24.5 |
| Air average humidity (%) | 63 | 62 | 62 | 66 | 64 | 64 |
| Maximum solar insolation (hours/day) | 7 | 8 | 8 | 8 | 8 | 8 |
| Average wind speed (km/h) | 14 | 13 | 13 | 11 | 10 | 10 |

ST: septic tank septage; PT: public toilet septage.

The results, in terms of the sludge moisture content (TH_Real) evolution along the six tests and water losses due to gravitational drainage (TH_Drenagem), are presented in Figure 6. TH_Drenagem data were directly measured considering the discharged volumes.

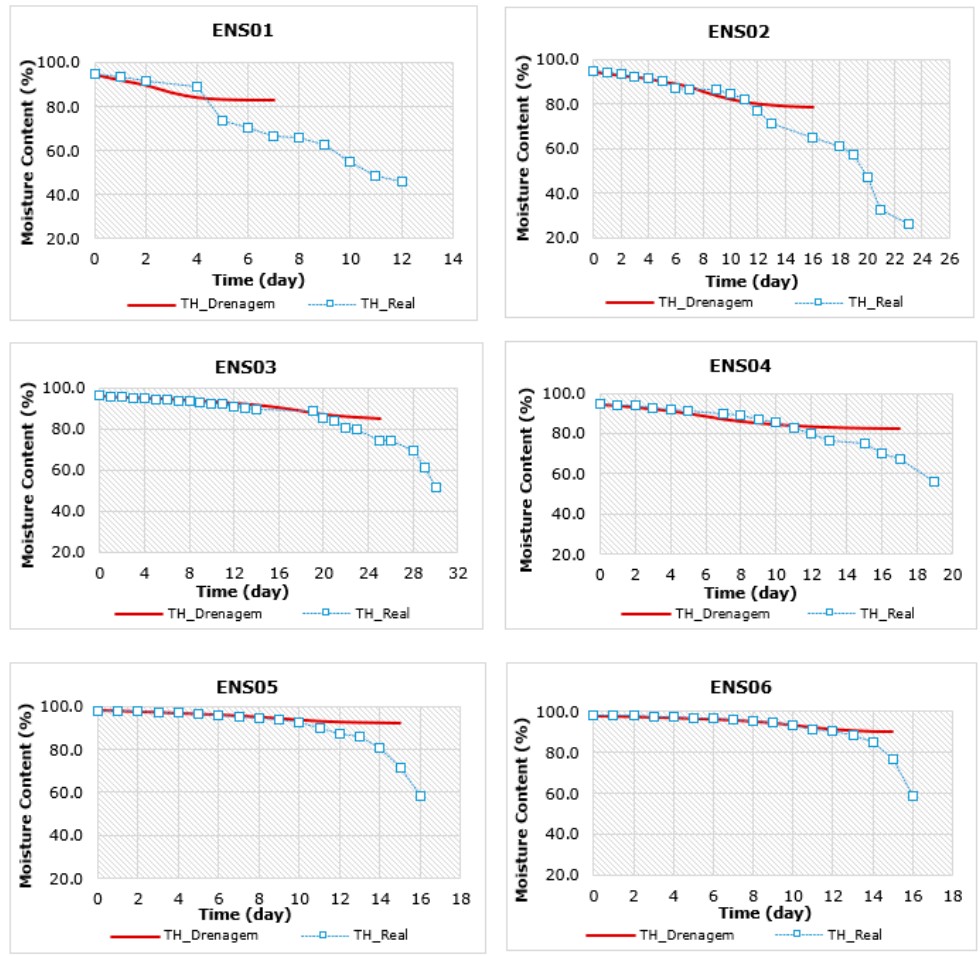

**Figure 6.** Sludge moisture content evolution (TH_Real) and water losses due to drainage (TH_Drenagem) for tests ENS01 to ENS06.

## 4. Discussion of the Experimental Results

### 4.1. Sludge Field Capacity

In the experimental campaigns, it was found that the sludge field capacity varied between 94 and 82%, with negligible amounts of water removed by gravitational drainage below 82% humidity. This result was expected as most authors [28] define a maximum value of up to 80% humidity as the limit for the occurrence of gravitational drainage. When adjusting the parameters of Equation (8) to the obtained results, a $k_{C_c}$ of 129.9 kgm$^{-3}$ and a $\alpha_{C_c}$ of 0.118 were obtained.

### 4.2. Total Drainage Time

The results of the total drainage time in the function of the initial depth of the sludge layer (modelled by Equation (12)) are presented in Figure 7. The experimental results indicate that the increase in the initial sludge depth resulted, as expected, in the increasing of the total drainage time and that this increase varied linearly. The obtained values for parameters $k_{t_d}$ and $\alpha_{t_d}$ were 55.6 m/s and 1.13, respectively.

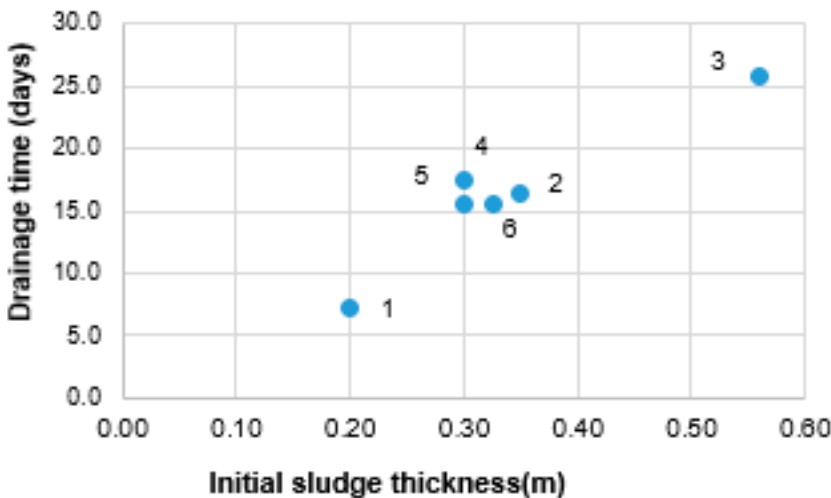

**Figure 7.** Initial sludge depth (between 0.20 and 0.55 m) versus drainage time.

### 4.3. Filter Medium Resistance

A filter medium resistance ($R_m$) was estimated from Equation (11) based on the results of the hydraulic conductivity tests carried out, as described in Section 2.2. The value found corresponded with about 1% of the total resistance.

It is well known that the filter medium is characterised by a high permeability, which is the reason why many authors (e.g., [21,23]) do not consider $R_m$ in the proposed models.

### 4.4. Cake Specific Resistance

The cake specific resistance was determined using Equation (9). As previously mentioned, $\alpha$ is a parameter that varies throughout the evolution process of the cake, largely due to compressibility, and it is common to use an average value to describe the specific resistance. To determine the $\alpha$ value for Equation (10) that best fit the dataset, the least squares method was used. A $\alpha$ between 2.3 E10 and 6.6 E10 m/kg was obtained. These results are comparable to those expected for digested or partially digested sludge, being similar to the values determined by [10,21].

### 4.5. Mass Transfer Coefficient

The mass transfer coefficient ($K_x$) was estimated by Equation (14) considering the SDB operational characteristics and local weather parameters, namely relative humidity, air temperature, sludge temperature and wind speed. It was found that the experimental values of $K_x$ varied in the range of 0.007 to 0.013 kg/s/m$^2$.

The local model parameters (mass transfer coefficient and cake specific resistance) are presented in Table 3.

**Table 3.** Local model parameters (mass transfer coefficient and cake specific resistance).

| Parameter | Unit | Determined Values |
|---|---|---|
| Mass transfer coefficient ($K_x$) | kg/s/m$^2$ | 0.007–0.013 |
| Cake specific resistance ($R_c + R_m$) | m/kg | $2.3$–$6.6 \times 10^{10}$ |

After the local parameters were experimentally determined, the model was used to compare the experimental results with the simulated results (Figure 8). In general, the progress of the curves shows an acceptable approximation, especially during the initial phase of the dewatering process. In the final phase, a deviation occurs in most tests, resulting (in general) in sludge moisture contents lower than those experimentally verified.

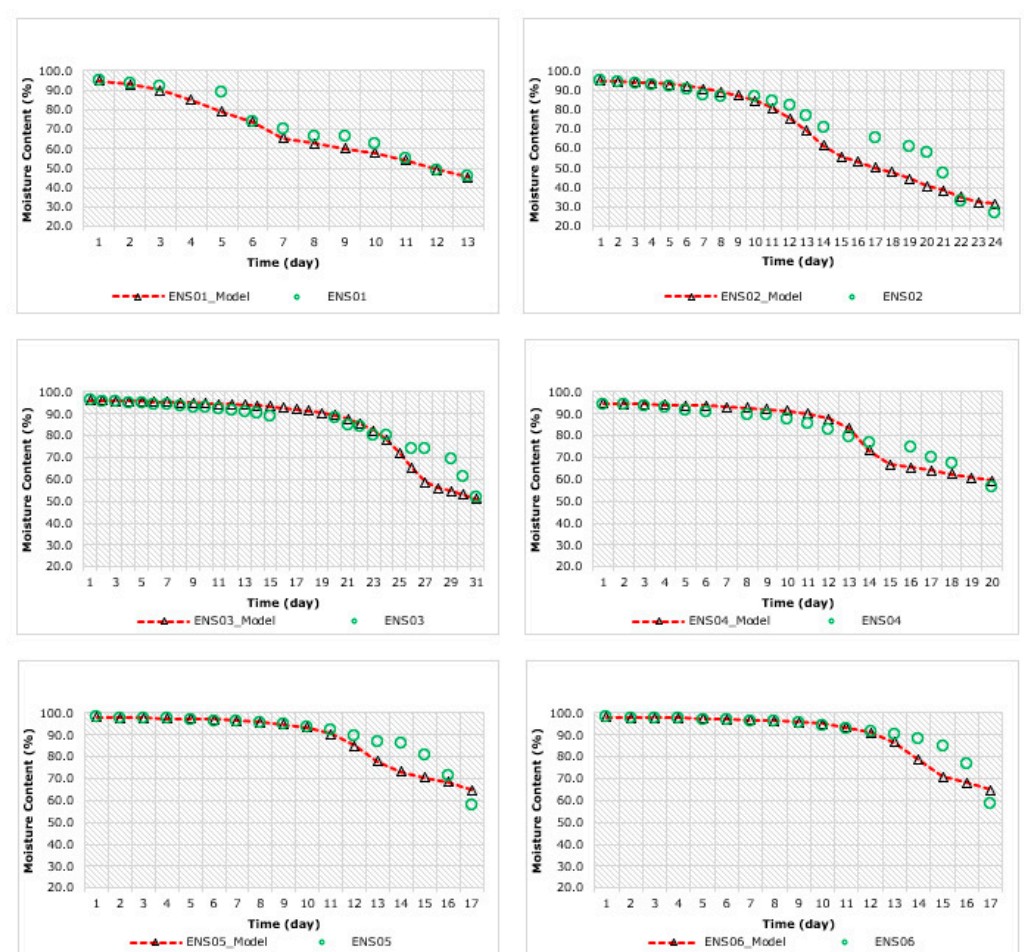

**Figure 8.** Comparison between the experimental results and the simulation results (tests EN01 to EN06).

To evaluate the reliability of the model, a statistical analysis between the results from the model and the experiment was conducted. The mean, standard deviation (SD), variance (Var) and number of observed/modelled points (n) for each test and for the global set of data are presented in Table 4.

**Table 4.** Mean, standard deviation, variance and number of observed/modelled points for each test and for the global set of data.

|  | ENS01 | | ENS02 | | ENS03 | | ENS04 | | ENS05 | | ENS06 | | Global | |
|---|---|---|---|---|---|---|---|---|---|---|---|---|---|---|
|  | Exp. | Model | Exp. | Model | Exp. | Model | Exp. | Model | Exp. | Model | Exp. | Model | Exp. | Model |
| Mean | 71.5 | 69.7 | 74.9 | 66.8 | 85.9 | 84.0 | 83.4 | 82.0 | 90.0 | 88.4 | 91.3 | 89.9 | 83.4 | 81.5 |
| SD | 17.5 | 17.2 | 21.0 | 24.9 | 11.7 | 15.7 | 11.1 | 14.5 | 10.9 | 12.0 | 10.2 | 11.4 | 15.4 | 18.1 |
| Var | 304.6 | 295.5 | 442.8 | 619.3 | 137.8 | 245.9 | 122.3 | 211.2 | 119.1 | 143.8 | 103.1 | 130.4 | 237.4 | 328.6 |
| n | 12 | 13 | 19 | 24 | 25 | 31 | 17 | 20 | 17 | 17 | 17 | 17 | 107 | 107 |

A Bland–Altman analysis was also carried out between the results from the model and the experiments, as shown in Figure 9. Overall, the Bland–Altman plot illustrates the good agreement between the experiments and the model. However, the mean is offset, lying above zero, suggesting a mean bias. No data lie below the lower limit of agreement (assuming a 95% limit of agreement for each comparison, i.e., an average difference $\pm$ 1.96 standard deviation of the difference) and 5.6% of the data lie above the upper confidence bound, suggesting a skew in the data.

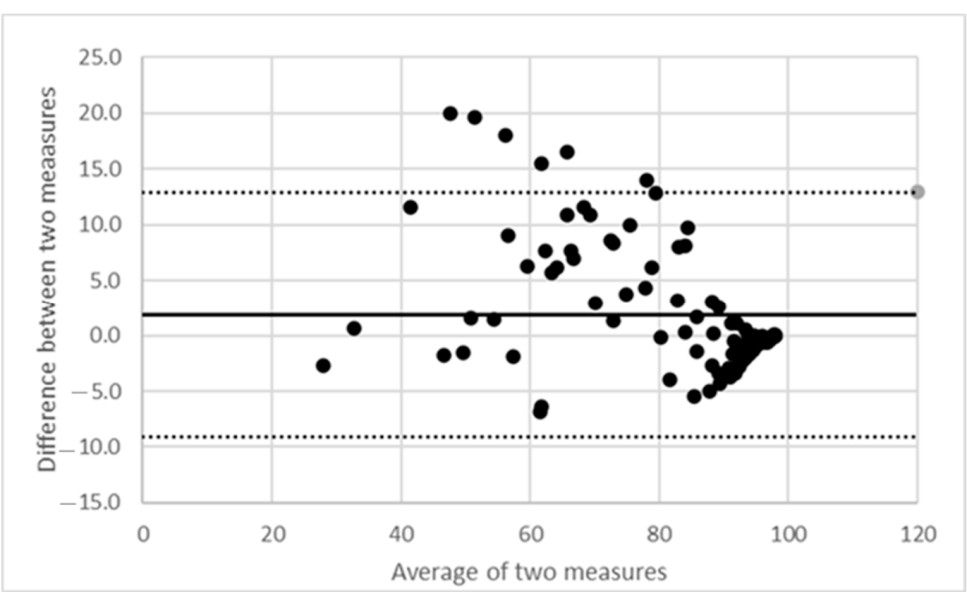

**Figure 9.** Bland–Altman plot analysing the results from the model and the experiment.

Although in most of the tests, particularly for the first 10 days, the model results were much closer to the field results, there were several relevant differences particularly in tests 1 and 2. In these two tests, the Student's t-test did not allow us to conclude, with 95% confidence, that there was no significant difference between the measured results and the values obtained by the model. A possible explanation may lie in the simplicity of the model that did not take into account, in a complete way, the nature of the sludge. It should be noted that in the experiments, the sludge came from septic tanks from households and from public toilets with different stabilisation levels.

## 5. Conclusions

In this paper, the experimental results of the pilot units and the results of a mathematical model describing the hydraulic performance of faecal sludge drying beds were presented and discussed. Local model parameters were determined based on the experimental data obtained from sludge dewatering pilot units located in Tete, central Mozambique, where typical warm dry weather conditions prevail.

A general acceptable agreement was verified between the predicted results and the measured ones. The sludge moisture content evolution in the six experimental tests for septage from two different sources, household septic tanks and public toilet septic tanks, was satisfactorily simulated and included both drainage and evaporation processes in the first 10 days of the dewatering process. After the 10 days, the deviations between the model results and the field results were generally observed, although the differences were not substantial. For future work, instead of simply considering $K_d$ (the empirical constant reflecting the resistance to the water flow) in Equation (6), two empirical parameters, $K_{d1}$ and $K_{d2}$, may be considered. These would reflect, respectively, the cake resistance during the intermediate phase and the cake resistance during the final drainage phase (being $t_{d1}$ and $t_{d2}$, respectively, the time of completing the intermediate phase and the time to complete the drainage process). The field studies should be planned considering those purposes.

The results suggest that this type of model may be used to estimate, in similar situations, the faecal sludge dewatering process in SDBs once the local key parameters are experimentally obtained. For different locations, given the dependency of the dewatering processes on-site climate variables and sludge characteristics, it is recommended that specific model constants, namely, the total drainage time, the mass transfer coefficient and the cake specific resistance, are determined.

Given the fact that in most of the sub-Saharan African developing countries, the urban population is typically served by on-site sanitation facilities, the present research might be considered to be a contribution to SDB planning and operation, allowing the estimation of drying times in the function of the initial sludge thickness and taking into account the final moisture content requirements.

**Author Contributions:** O.M.: literature review, experimental studies and model development and writing—original draft preparation. R.M.: contribution to model development and analysis. F.F.: conceptualisation and writing of the final paper. J.M.: conceptualisation and supervision. All authors have read and agreed to the published version of the manuscript.

**Funding:** This research was funded by Camões—Instituto da Cooperação e da Língua, I.P. and by ARA-Zambeze.

**Institutional Review Board Statement:** Not applicable.

**Informed Consent Statement:** Not applicable.

**Data Availability Statement:** Data supporting reeported results can be found in: Moiambo, O. Simulação Hidráulica da Desidratação de Lamas Fecais, Baseada em Estudos de Caso em Portugal e Moçambique. Ph.D. Thesis, Instituto Superior Técnico, Universidade de Lisboa, Lisbon, Portugal, 2018.

**Acknowledgments:** The authors express their gratitude to the following institutions: (i) Camões, Instituto da Cooperação e da Língua, CICL, for the funding of the PhD in Environmental Engineering at the Instituto Superior Técnico of the Lisbon; (ii) to the Administração Regional das Águas do Zambeze, ARA-Zambeze, for financing the assembly and operation of the pilot treatment plant faecal sludge and (iii) to the Conselho Municipal da Cidade de Tete, for logistical support in the collection of the sludge and the loading of the drying beds.

**Conflicts of Interest:** The authors declare no conflict of interest.

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
