# Peer review of "Modelling Faecal Sludge Dewatering Processes in Drying Beds Based on the Results from Tete, Mozambique"

_sustainability, doi:10.3390/su13168981_

Round 1

Reviewer 1 Report

Manuscript ID sustainability-1285524 entitled “Modeling faecal sludge dewatering processes in drying beds based on results from Tete city, Mozambique” presented a mathematical model for faecal sludge dewatering in sludge drying beds (SDB), focusing on gravity drainage and evaporation processes, is presented and discussed . In my opinion, the presented “technology” for dewatering sludge with different characteristics in natural conditions has been widely known for many years. Models describing the drainage mechanisms have already been developed and presented several dozen years ago.

I believe that paying attention to primitive manure management is unjustified and unnecessary. In my opinion, the manuscript does not add anything new to the commonly known knowledge. The modeling process performed is very simple and does not provide any new knowledge in this field.

It should be emphasized that sludge dewatering under natural conditions was also practiced in European countries, and at present, due to the applicable legal regulations, the need to reduce the emission of gases, aerosols, odors and for sanitary reasons has been abandoned.

In my opinion, the subject of the manuscript is wrong.

Reviewer 2 Report

This paper provides a model and its experimental validation for the fecal sludge dewatering process.

Overall, I found this study interesting and informative. I am rating a few questions and comments for the authors for (my) better understanding.

1. I feel that some important parameters are being missed in the provided model.
One is pressure. The pressure not only is effective on the mass transfer coefficient (as described by the authors), but also it is effective for the physical dewatering process. To make it easy, I want to ask, what will happen if you connect the drainage tank to a compressor.
Of course, my understanding from figure 3 is that the system is open and the only pressure over the cake will be free air and water. That might be a reason for neglecting the pressure in the model. However, for generalizing the model, it is essential to consider it.
Another important parameter is the filter media. How did you characterize it? different types of filters with different opening sizes will even affect the shape of the case. Please explain more about the Kd. How did you measure it?
Another parameter is the chemical characteristics of the sludge, such as water content and the size of the particles.

I invite the authors to discuss how their model can consider these essential parameters.

2. What are the boundary conditions of the model? Please discuss this clearly.

3. Figure 8 is very interesting. How the model can be so predictable, without considering the characters that I have described in item 1 of my comments? I invite the authors to defend the reliability of their model by providing:
A. a sufficient statistical analysis between the results from model and experiment. I suggest testing the differences between values given by the model and the measured ones from the experiment through both t-test and Bland-Altman analysis.
B. an understandable sensitivity analysis for all parameters.

4. Please describe the filter medium in your experiment and clearly explain how Kd was measured for them.

Reviewer 3 Report

The authors conducted a pilot experiment to validate their model of the dewatering process in sludge drying beds. The mathematical model is well defined and the experiment was well fitted to the model prediction. The manuscript is well written with validated data and results.

Author Response

Reviewer 3 found the study excellent and ready to publication and as so did not present suggestions for improvement.

Round 2

Reviewer 1 Report

Thanks for Authors response and explanation. Manuscript has been also improved agree with commnents other Reviewers. In my opinion it can be publish in present form.

Author Response

Thanks for Authors response and explanation. Manuscript has been also improved agree with comments other Reviewers. In my opinion it can be publish in present form.

We are grateful for the reviewer comment and very pleased that the paper is considered appropriate to be published.

Reviewer 2 Report

I have checked the revised version of the manuscript, and the authors' reply.

Excellent work! The manuscript has been significantly improved. Overall, considering the results for statistical analysis to investigate the differences between the real data and model, I think the model is valid enough for its purpose. Congratulations!

I only have one comment and one question, which I wish to discuss with the authors. The authors are welcomed to discuss them in their paper if they wish.

Comment: Please provide references to validate basic equations, especially for Equation 30.

Question: My last concern about the model is that although the authors have considered the filter's characteristics in their model, it should be noted that the filter's characteristics will certainly change with time as it gets clogged. Moreover, when the sludge cake forms, it will also resist against dewatering process and acts like a filter itself. This may be a reason that the author's model's prediction is robust for the first 10 days. So, what is your opinion about this, and how your model can be revised for this issue (in this study or further studies)? Your work can be perfect if you may consider discussing this matter.
